

# Quality-of-life among Syrian refugees residing outside camps in Jordan relative to Jordanians and other countries

Nour Abdo[1], Faten Sweidan[2] and Anwar Batieha[1]

[1] Department of Public Health and Family Medicine/Faculty of Medicine, Jordan University of Science and Technology, Irbid, Jordan
[2] Department of Public Health/School of Medicine, Jordan University of Science and Technology, Irbid, Jordan

## ABSTRACT

**Background:** Since the beginning of the Syrian humanitarian crisis, Syrians sought refuge in many safer countries. Many aspects of Syrian refugees' lives have been affected, hence affecting the overall quality of their lives. However, only one study has investigated their quality of life (QOL). The aim of this study was to assess the QOL of Syrian refugees residing outside camps in Jordan and compare it to the QOL of Jordanians and to other refugees and populations around the globe.

**Methods:** Data were obtained from Syrian refugees residing outside camps in Jordan, and from two Jordanian groups; low socioeconomic status (LSES) Jordanians and average socioeconomic status (ASES) Jordanians in 2017. A total of 661 Syrians, 208 LSES Jordanians and 714 ASES Jordanians, aged between 18 and 75 years were included. The World Health Organization Quality of Life instrument (WHOQOL-BREF) questionnaire was used as the QOL assessment tool. Analysis of Variance "ANOVA" and post hoc Tukey-Honest tests were used to find the differences between the means of QOL questions in the three groups (Syrians, ASES, and LSES). Stepwise multivariate linear regression was performed for each domain to determine the most associated risk factors.

**Results:** No significant difference was found between Syrian refugees and LSES Jordanians in the physical health domain. Syrian refugees scored significantly lower than LSES Jordanians in the psychological health and social relationship domain. Syrian refugees scored significantly higher than LSES Jordanians in the environmental domain. ASES Jordanians scored significantly higher than the other two groups in all domains, with all its scores above the average.

**Discussion:** Despite the support Jordan provides to the Syrian refugees, they still seem to suffer from poor psychological health, social relationships and environmental domains, with scores below 50 on (0–100) scale. Nonetheless, no significant difference was found between Syrian refugees and LSES Jordanians in the physical health domain, furthermore they scored significantly higher than LSES Jordanians in the environmental domain despite both scoring below 50 on (0–100) scale in this domain. Physical, psychological, and social domains were mainly affected by having a job, having higher income, and being married and free from diseases.

Corresponding author
Nour Abdo, nmabdo@just.edu.jo

## INTRODUCTION

Since the beginning of the civil war in Syria, more than five million Syrians are registered as refugees. Around three million Syrian refugees are registered in Turkey and the remaining two million are mainly (80%) registered in Lebanon and Jordan. The rest are registered in Iraq and Egypt. Almost 19% of the registered refugees in Jordan are residing in camps (*United Nations High Commissioner for Refugees (UNHCR), 2018*). Jordan has registered more than 673,000 Syrian refugees in the UNHCR databases (*United Nations High Commissioner for Refugees (UNHCR), 2018*). According to the UNHCR database, five camps are available for Syrian refugees in Jordan, and the main three camps are Al-Zaatari camp with 78,545, Al-Azraq camp with 40,738, and the Emirati Jordanian camp with 6,848 refugees. The remaining reside outside camps (*United Nations High Commissioner for Refugees (UNHCR), 2017b*). Camps are the first stop for refugees without identification documents, where they have the choice to stay and benefit from the privileges the camp has to offer or leave to depart from the camp, and be responsible for providing food, rent, medical care, and education for their families (*United Nations High Commissioner for Refugees (UNHCR), 2017c*).

Quality of Life (QOL) can be defined as the general well-being of individuals and societies, outlining negative and positive features of life. It observes life satisfaction, including everything from physical health, family, education, employment, wealth, religious beliefs, finance, and the environment (*School, 2013*). Forced displacement of Syrians affected their QOL, including its psychological (*Weinstein, Khabbaz & Legate, 2016*), physical (*Doocy et al., 2016*), and social aspects (*Sevinç et al., 2016*). Syrian refugees have suffered several psychological consequences. The literature has shown that Syrian refugees are struggling with deep pain and distress due to poverty, job insecurity, and unemployment (*Anagnostopoulos, Giannakopoulos & Christodoulou, 2016*). Furthermore, many of them were diagnosed with depression, generalized stress and post-traumatic stress disorder (*Weinstein, Khabbaz & Legate, 2016*). On the other hand, the physical health of Syrian refugees was also distraught. Due to the destruction of hospitals and chaos as a result of the Syrian civil war, severe disruptions to health care services were observed (*Ozaras et al., 2016*).

Consequently, several infectious diseases have reemerged (*Ozaras et al., 2016*), and an increase in the prevalence of uncontrolled non-communicable diseases was observed (*Gammouh et al., 2015*). The nutritional status of the refugees was also a field of study, as literature shows high prevalence of anemia among non-pregnant Syrian women and children in the camp settings (*Bilukha et al., 2014*).

Our study was the first to assess the QOL of Syrian refugees residing outside camps. There was only one study that focused on the QOL of Syrian refugees, and was conducted on the Syrian refugees' camp population in Kurdistan (*Aziz, Hutchinson & Maltby, 2014*). No studies were found concerning the QOL of Syrian refugees residing outside

camps. This study is a great opportunity to illuminate our understanding on how Syrian refugees outside camps are coping with their new environment.

Furthermore, the study provided a snapshot estimation of QOL among two Jordanian groups (middle income Jordanians and Jordanians with low-socioeconomic status). Moreover, the results of this research will help the government as well as external and internal donors to properly allocate their financial and medical assistance.

This study assessed the QOL of registered refugees, therefore, it was not representative of unregistered refugees, who were unable to get access to healthcare the way registered Syrian refugees do; similarly the studied population were urban area residents who had no access to the free accommodation, health services, food and education privileges that camp residents have.

## MATERIALS AND METHODS

### Sample and sampling procedure

The following three groups were recruited: Syrian refugees, average socioeconomic status (ASES) Jordanians, and low socioeconomic status (LSES) Jordanians. A total sample of 1,583 participants was collected as detailed below. It included everyone willing to participate aged 18 years old or above. Data collection took place between February and August 2017.

### Syrian refugees

Syrian refugees was recruited from the refugees and their companions (if any), who sought the Caritas Al-Husn Center for Humanitarian and Medical Services between February and June 2017. The Caritas Al-Husn Center serves Syrian refugees residing outside camps as well as vulnerable Jordanians. Caritas Al-Husn Center refers some of the beneficiaries to a local pharmacy, so we used the pharmacy as a data collection point when data collection became hard to accomplish at the center when no proper space was available for data collection anymore due to the increase in beneficiary numbers. Data collection was also performed at two humanitarian assistance campaigns for the Syrian refugees; they were designed to provide food, and took place at Al-Hasan Sports City-Irbid during the data collection period.

### Average socioeconomic status Jordanians

Average socioeconomic status Jordanians were defined as people with average living circumstances, and not eligible for social development department (SDD) financial assistance. This category was recruited from Al-Husn Health Care Center from February to August 2017. Everyone who attended the clinic in the specified period and who were willing to participate was included in the study.

### Low socioeconomic status Jordanians

Low socioeconomic status Jordanians were defined as people who were eligible for services and assistance from the SDD. The SDD has a strict vetting system to identify their eligibility. The SDD conducts a social vulnerability study for the applicants who are seeking financial assistance, and requests a number of documents: bank statements showing their monthly salary, loans, and monthly deductions, prison services certificates,

detailed medical reports explaining the reason why the family should get financial assistance from SDD, detailed medical reports for any disabled family members, divorce and custody certificates, alongside their identity documents including their family book and identification cards (*Development MoS, 2018*). We recruited LSES Jordanians from SDD in Irbid. Anyone who attended the SDD from February to May 2017 and who were willing to participate was included in the study.

## DATA COLLECTION

### Measures

WHOQOL-BREF is a QOL assessment tool, developed by the *World Health Organization (WHO) (2012)*. WHOQOL-BREF is the short version of a much more detailed questionnaire, WHOQOL-100, and can be used as an efficient alternative in studies aiming to assess QOL. WHOQOL-BREF is much more practical to use in surveys than WHOQOL-100 as using 26 questions to assess QOL is easier than using 100 questions in the original version. It's rapid, easy and gives a good estimate of the overall QOL. WHOQOL-BREF assesses the quality of physical health (seven items; e.g., "To what extent do you feel that physical pain prevents you from doing what you need to do?"), psychological health (six items; e.g., "How satisfied are you with yourself?," social QOL (three items; e.g., "How satisfied are you with the support you get from your friends?"), and environmental QOL (eight items; e.g., "How healthy is your physical environment?") (*World Health Organization (WHO), 2012*).

Answers were scored from one to five with various anchor statements (e.g., from (Very dissatisfied) to (Very satisfied) or (Very poor) to (Very good)), except for question 26 that asked about the frequency the subject experiences negative feelings. Question 26's anchor statements had different meanings; (1: Never), (2: Seldom), (3: Quite often), (4: Very often), and (5: Always). The psychometric properties of the WHOQOL-BREF's have been validated as a QOL assessment tool for various cultures and socioeconomic status (*Skevington et al., 2004*).

### Data management and statistical analysis

A pilot study was performed to insure internal consistency. The test was performed on 56 participants, above 18 years of age and willing to participate from Al-Farouq Health Center in the Irbid area. Cronbach's alpha was above 0.7 for all domains except for the social relationships domain, and according to the WHO manual, the internal consistency test for this domain cannot be trusted because it consists of three items only, so we proceeded with the actual study. Data entry and analysis was performed using SPSS-PC software v20 and SAS v9.2 software. Data cleaning was performed to check for data entry errors; we performed range and logic checks to find out possible errors in data entry. Detected errors were corrected by returning back to the study forms, remaining errors were treated as missing if they were not possible to correct.

The assessment is discarded when more than 20% of the data is missing. In the case of a missing item, the mean of other items in the domain is substituted. When two items are missing, the domain score was not calculated according to WHOQOL-BREF

instructions, except for domain 3, where the domain 3 should only be calculated if $\leq 1$ items is missing. The Arabic version of the WHOQOL-BREF reliability and validity has been tested through a study done among large Arabic-speaking samples (*Ohaeri & Awadalla, 2009*). The following item of the social relationships QOL ("How satisfied are you with your sex life?") was only answered by 4.4% Syrians, 7.7% LSES Jordanians and 22.2% ASES Jordanians, due to the sensitivity of the question in this cultural context, although some participants did answer it. According to the WHOQOL-BREF manual, missing items are allowed in the transformational methods for scoring of the scale. ANOVA and post hoc Tukey-Honest tests were used to find the differences between the means of QOL questions in the three groups (Syrians, ASES, LSES). Stepwise multivariate linear regression was performed for each domain to determine the most associated factors. Population was included in all regression models as "population" is the variable of interest in this study. In addition to population (ASES Jordanians LSES Jordanians and Syrian refugees), adjustment for the following possible confounders was made: age, having a chronic disease, family size, monthly income, having a job, personal description of financial situation, education, marital status, and gender.

## Ethical considerations

The study protocol was approved from the institutional review board (IRB) of Jordan University of Science and Technology IRB 33/107/2017. All data were kept strictly confidential and used only for scientific reasons without identifying information for the participants. Verbal informed consent was obtained from each participant. The study carries no foreseeable harm to participants as it was based on interviewing the participants without any invasive procedures. Each person was given the choice of participating or not without any pressure.

# RESULTS

We were able to recruit 1,583 participants: 661 Syrians, 714 ASES Jordanians, and 208 LSES Jordanians. Data collection took place in Irbid area. The participants were residents of Irbid, Jarash, Ajloun, AlMafraq, Amman, and Al Salt.

Eight questionnaires were discarded because they had 20% or more missing data; the remaining 1,575 questionnaires were eligible for analysis: Syrians 655 (41.6%), LSES Jordanians 208 (13.2%), and ASES Jordanians 712 (45.2%). General demographics of the three populations are listed in Table 1.

## Domain results

### Physical health domain

Table 2 and Fig. 1A shows the means of physical health domain questions for the three populations. Syrian refugees scored average or less in all the questions and were the lowest among all groups. LSES Jordanians had the highest pain tolerance among all groups (F3) among all groups, followed by Syrian refugees then ASES Jordanians. Similarly, LSES Jordanians needed less medical treatment (F4) than the other two groups, followed by Syrian refugees and ASES Jordanians. Stepwise multivariate linear regression resulted

| Descriptives | Population N (%) | | |
|---|---|---|---|
| | Syrians 655 (41.6) | LSES Jordanians 208 (13.2) | ASES Jordanians 712 (45.2) |
| Gender | | | |
| Male | 266 (40.6) | 57 (27.4) | 116 (16.3) |
| Female | 389 (59.4) | 151 (72.6) | 595 (83.7) |
| Age | | | |
| 18–29 Years | 109 (17.0) | 34 (17.2) | 158 (23.1) |
| 30–39 Years | 202 (31.4) | 52 (26.3) | 193 (28.2) |
| 40–49 Years | 210 (32.7) | 66 (33.3) | 157 (23.0) |
| 50–59 Years | 88 (13.7) | 28 (14.1) | 112 (16.4) |
| 60 Years and above | 34 (5.7) | 18 (9.1) | 64 (9.4) |
| Education | | | |
| Illiterate | 46 (7.03) | 1 (0.5) | 32 (4.5) |
| Primary education | 451 (69.0) | 81 (39.0) | 115 (16.2) |
| Secondary education | 98 (15.0) | 83 (39.9) | 222 (31.3) |
| College education | 59 (9.0) | 43 (20.7) | 341 (48.0) |
| Marital status | | | |
| Single | 22 (3.4) | 26 (12.6) | 54 (7.6) |
| Married | 584 (89.4) | 140 (68.0) | 595 (83.6) |
| Separated | 11 (1.7) | 25 (12.1) | 27 (3.8) |
| Widow | 36 (5.5) | 15 (7.3) | 36 (5.1) |
| Family size | | | |
| Small (<5 members) | 186 (28.9) | 96 (47.1) | 321 (46.0) |
| medium (5–7 members) | 321 (49.9) | 81 (39.7) | 328 (47.0) |
| Large (>7 members) | 136 (21.2) | 27 (13.2) | 49 (7.0) |
| Job | | | |
| Yes | 81 (12.6) | 34 (20.9) | 230 (33.0) |
| No | 560 (87.4) | 129 (79.1) | 467 (67.0) |
| Monthly income (Per family JOD) | | | |
| Low income <300 | 546 (89.4) | 137 (74.9) | 213 (31.1) |
| Middle income 300–700 | 63 (10.3) | 44 (24.0) | 378 (55.1) |
| High income > 700 | 2 (0.3) | 2 (1.1) | 95 (13.9) |
| Personal description of financial situation | | | |
| Below Average | 505 (84.6) | 92 (60.1) | 156 (22.4) |
| Average and above | 92 (15.4) | 61 (39.9) | 540 (77.6) |
| Presence of disease | | | |
| Yes | 393 (60.2) | 94 (45.2) | 196 (27.5) |
| No | 260 (39.8) | 114 (54.8) | 516 (72.5) |

**Table 1 General characteristics of the studied groups, Jordan 2017.**

**Table 2 Means of domain's questions for the three populations, Jordan 2017.**

| Domain's questions | Population/Means[*] | | | |
|---|---|---|---|---|
| | Syrian population (N = 655) | LSES Jordanians (N = 208) | ASES Jordanians (N = 712) | P-value |
| Physical domain | | | | |
| f3: Physical pain | 2.7 | 2.87 | 2.38 | <0.0001 |
| f4: Need for medical treatment | 2.69 | 2.69 | 1.99 | <0.0001 |
| f10: Energy | 2.81 | 2.8 | 3.25 | <0.0001 |
| f15: Mobility | 2.98 | 3.08 | 3.74 | <0.0001 |
| f16: Sleep | 3.02 | 2.93 | 3.48 | <0.0001 |
| f17: Satisfaction of the daily activities performance | 2.9 | 3.13 | 3.55 | <0.0001 |
| f18: Satisfaction of the capacity of work | 2.87 | 3.14 | 3.62 | <0.0001 |
| Psychological domain | | | | |
| f5: Enjoying life | 2.68 | 2.43 | 3.07 | <0.0001 |
| f6: Meaningful life | 2.93 | 2.79 | 3.45 | <0.0001 |
| f7: Concentration ability | 2.89 | 3.07 | 3.24 | <0.0001 |
| f11: Acceptance of bodily appearance | 3.07 | 3.54 | 3.72 | <0.0001 |
| f19: Satisfaction of oneself | 3.29 | 3.49 | 3.82 | <0.0001 |
| f26: Negative feelings | 2.85 | 3.45 | 2.78 | <0.0001 |
| Social domain | | | | |
| f20: Relationships satisfaction | 3.22 | 3.64 | 3.9 | <0.0001 |
| f21: Sexual life satisfaction (29,16,158)[a] | 3.34 | 3.44 | 3.81 | 0.05 |
| f22: Satisfaction with friends support | 2.76 | 2.65 | 3.34 | <0.0001 |
| Environmental domain | | | | |
| f8: Feeling secure | 3.26 | 3.14 | 3.71 | <0.0001 |
| f9: Physical environment | 3.08 | 2.73 | 3.28 | <0.0001 |
| f12: Money availability | 2.34 | 1.93 | 2.94 | <0.0001 |
| f13: Information availability | 2.87 | 3.02 | 3.27 | <0.0001 |
| f14: Opportunity for leisure activities | 2.41 | 1.99 | 2.38 | <0.0001 |
| f23: Living place environment | 3.14 | 2.9 | 3.45 | <0.0001 |
| f24: Healthcare access satisfaction | 3.01 | 2.89 | 3.68 | <0.0001 |
| f25: Transportation | 3.04 | 3.06 | 3.67 | <0.0001 |

**Notes:**
[*] On the Likert scale were 5 means very satisfied or very good, and 1 means very dissatisfied or very poor.
[a] N(%) was 29(4.4%) for Syrian refugees, 16 (7.7%) for LSES Jordanians and 158 (22.2%) for ASES Jordanians.

in the following: overall physical health was better for ASES Jordanians, followed by LSES Jordanians then by the Syrian refugees. Aging, having a disease, being married and increasing family size had negative effect on overall physical health. On the other hand, higher monthly

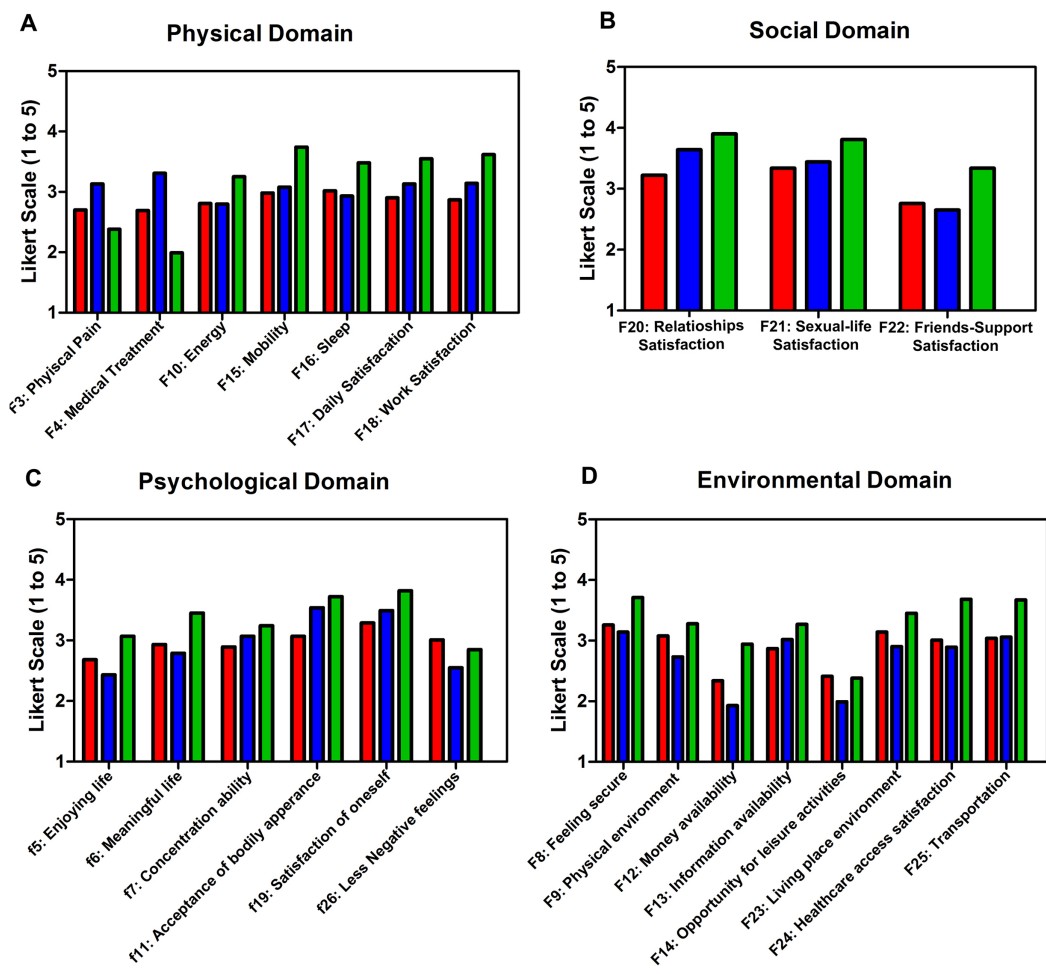

**Figure 1 Quality-of-life domain question means in Syrians refugees residing outside camps in Jordan and Jordanians.** (A) The mean score for each question in the Physical-Health domain for Syrian refugees (red), LSES Jordanians (blue), ASES (green). (B) The mean score for each question in the Social domain for Syrian refugees (red), LSES Jordanians (blue), ASES (green). (C) The mean score for each question in the Psychologcial-Health domain for Syrian refugees (red), LSES Jordanians (blue), ASES (green). (D) The means score for each question in the Environmental-Health domain for Syrian refugees (red), LSES Jordanians (blue), ASES (green).

income and having a job had a positive effect on overall physical health, Table 3. Supplemental Table shows analysis stratified for each population for each domain.

### Psychological health domain

Although Syrian refugees scored significantly lower than both Jordanian groups when asked about their concentration ability (F7), acceptance of bodily appearance (F11), and satisfaction of oneself (F19), they still scored around average. Syrian refugees are enjoying life (F5) more than LSES Jordanians, but they scored below average.

Jordanians of low socioeconomic status experienced negative feelings more than the other two groups, while Syrian refugees experienced lesser amounts of negative feelings compared to other groups, Table 2 and Fig. 1C. Stepwise multivariate linear regression resulted in the following: the overall psychological health was better for ASES Jordanians,

**Table 3 Stepwise multivariate linear regression for each domains.**

| Variables | Change in domain scores | P-value |
|---|---|---|
| **Physical health domain** | | |
| Intercept | 62.75 | <0.0001 |
| Population | −3.63 | <0.0001 |
| Age | −0.15 | <0.0001 |
| Diseases | −2.5 | <0.0001 |
| Family size | −0.54 | <0.05 |
| Monthly income | 0.01 | <0.0001 |
| Job | 3.93 | <0.0001 |
| Personal description of financial situation | 3.49 | <0.001 |
| Marital status | 1.22 | <0.05 |
| **Psychological health domain** | | |
| Intercept | 52.78 | <0.0001 |
| Population | −1.93 | <0.001 |
| Age | −0.11 | <0.001 |
| Diseases | −0.89 | <0.05 |
| Monthly income | 0.01 | <0.001 |
| Job | 2.21 | <0.05 |
| Personal description of financial situation | 6.98 | <0.0001 |
| Marital status | 1.1 | <0.05 |
| **Social relationships domain** | | |
| Intercept | 55.38 | <0.0001 |
| Population | −5.12 | <0.0001 |
| Diseases | −1.04 | <0.05 |
| Monthly income | 0.0 | <0.05 |
| Personal description of financial situation | 6.80 | <0.0001 |
| Marital status | 1.29 | <0.05 |
| **Environmental domain** | | |
| Intercept | 51.33 | <0.0001 |
| Population | −1.05 | <0.05 |
| Monthly income | 0.01 | <0.0001 |
| Education | −0.14 | >0.05 |
| Gender | −2.78 | <0.001 |
| Personal description of financial situation | 8.54 | <0.0001 |

followed by LSES Jordanians and Syrian refugees, respectively. Aging and disease had a negative effect on overall psychological health, while on the other hand higher monthly income and having a job had positive effect on overall physical health (Table 3; Supplemental Table).

### Social relationships domain
Average socioeconomic status Jordanians scored significantly higher than the other two groups in all questions in this domain. Syrian refugees scored significantly the lowest

**Table 4 Overall means scores for each domain in the three populations (0–100 scale).**

| Domain | Syrian population (N = 655) | LSES Jordanians (N = 208) | ASES Jordanians (N = 712) | P-value |
|---|---|---|---|---|
| Physical health | 50.68 | 48.68 | 65.28 | <0.0001 |
| Psychological health | 49.35 | 53.23 | 60.24 | <0.0001 |
| Social relationships | 49.82 | 53.59 | 65.89 | <0.0001 |
| Environmental domain | 47.37 | 42.67 | 57.39 | <0.0001 |

among all groups in the first two questions and higher than LSES Jordanians in the last question, Table 2 and Fig. 1B.

As mentioned above, a few answered Q21: ("How satisfied are you with your sex life?"), all who answered scored above the cut point in the three populations, with no significant differences between them.

After adjustment for the following factors: population (Syrian refugees, LSES, ASES), age, having a chronic diseases at the time of the study, family size, monthly income, having a job at the time of the study, personal description of financial situation, education, gender, and marital status, stepwise multivariate linear regression resulted in the following: the overall quality of social relationships was better for ASES Jordanians, followed by LSES Jordanians then Syrian refugees. Diseases had a negative effect on the overall quality of social relationships. High monthly income and being married had positive effect on the overall physical health (Table 3; Supplemental Table).

Syrian refugees had significantly healthier environment, were more satisfied with their living conditions (F23) and had significantly more money (F12) than LSES Jordanians (F9), Table 2 and Fig. 1D. Stepwise multivariate linear regression resulted in the following: the overall environment quality was better for ASES Jordanians, followed by LSES Jordanians and Syrian refugees respectively. Higher monthly income had positive effect on the overall physical health. Males were more satisfied with their environment quality (Table 3; Supplemental Table).

### General look at the QOL domains for each population

Syrian refugees scored significantly higher than LSES Jordanians in the environmental domain but were still below average; on the other hand they scored lower than LSES Jordanians in the psychological health and social relationship domain. No significant difference was found between Syrian refugees and LSES Jordanians in the physical health domain. ASES Jordanians scored the highest among all groups in all domains, with all its scores above the average, Table 4 and Fig. 2.

## DISCUSSION

Since the beginning of the Syrian humanitarian crisis, Jordanian government has been trying to provide needed care for Syrian refugees. Over the past two decades, Jordan has hosted a lot of refugees from different countries due to political unrest in the area. After the influx of Syrian refugees to Jordan, more than 271 million Jordan Dinars (JODs) were needed in public health facilities, out of the 1.5 billion JODs that have been spent on

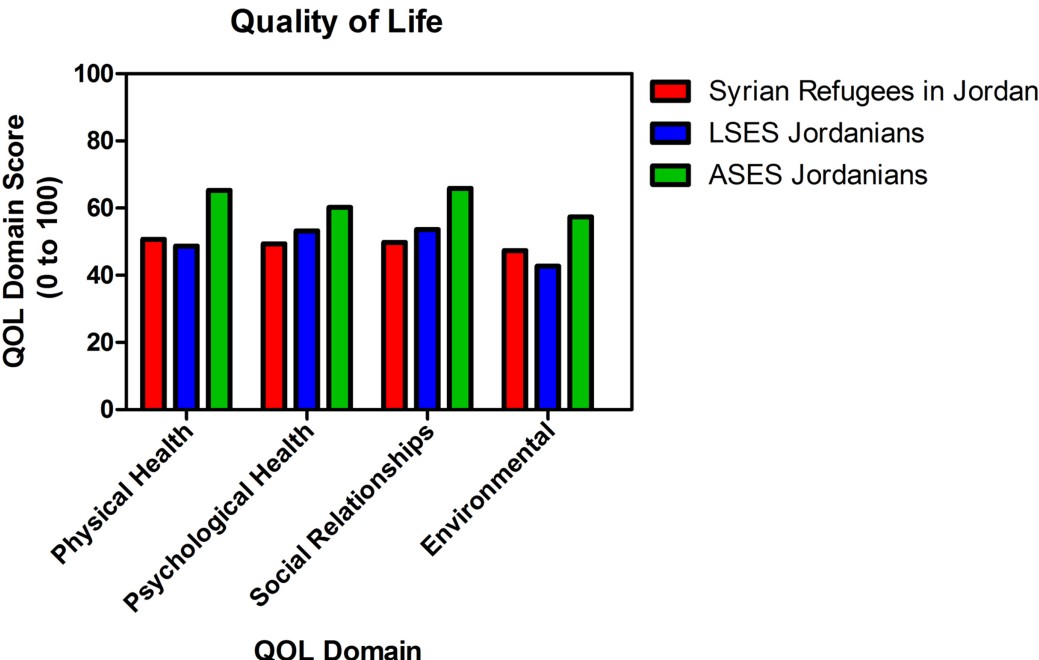

**Figure 2 Overall Quality of Life means in Syrians refugees residing outside camps in Jordan and Jordanians.** Mean score for each domain for Syrian refugees, LSES and ASES Jordaninans.

healthcare services by the end of 2016, and the demand for healthcare services is growing (*Al Emam, 2016*). Since the beginning of the crisis, more than 673,000 Syrians had registered as refugees. However, the actual number is around 1.4 million refugees and this creates a load on healthcare services. In the time period between 2012 and 2016, Ministry of Health hospitals provided services to around 630,000 Syrian refugees, with around 59,000 hospital admissions. Healthcare centers across the country received around 11 million Syrian refugees' visits (*Al Emam, 2016*). UNHCR is also providing primary, secondary and tertiary healthcare services free of charge for vulnerable Syrians (UNHCR). We assessed in this study the QOL in a large sample of Syrian refugees, and compared it to LSES and ASES Jordanians and refugees worldwide. No studies assessed the QOL for Syrian refugees outside camps. However, one study assessed the QOL for in-camp Syrian refugees (*Aziz, Hutchinson & Maltby, 2014*), and other studies assessed the QOL for other refugees and groups (*Akinyemi et al., 2012*).

## Physical health
Syrian refugees had significantly lower physical pain and need for treatment than the ASES Jordanians, but significantly higher than LSES Jordanians. Syrian refugees also scored more or less the same as the LSES Jordanians scores in most of the physical health domain questions. The overall physical health score for the Syrian refugees were very close to LSES Jordanians and the scores were around average, in opposite to ASES Jordanians who scored above average. Syrian refugees are still facing challenges in obtaining healthcare services despite the assistance Jordan is trying to provide, either because

of the cost, especially after the new healthcare fees imposed in 2014, lengthy bureaucratic procedures, or the lack of the documents needed for healthcare access eligibility.
At least 58.3% of Syrian adults with chronic conditions are not able access health services including medicines, according to the UNHCR (*Amnesty International, 2016*).

## Psychological health

The psychological scores were below average for most of the questions, and the lowest among all three groups in the overall questions, those findings are maybe due to the challenges that any refugee goes through trying to build new life in different country. In the first 2 years of this crisis, The Ministry of Health spent about US $53 million on care to refugees, with only US $5 million provided by UN agencies. That being said, proper mental health care seems challenging. Finance is just one of many challenges; a lack of trained mental health practitioners, and the lack of the basic factors essential to mental health (that is, education for children, employment for adults, comfortable and a sanitary living environment) are not yet accessible to many Syrian refugees in Jordan (*Al Hadid, 2016*).

## Social relationships

Syrian refugees scored significantly less than the other two groups in the overall social relationships domain, and they were not satisfied with the support they got from their friends, maybe due to the hard economical circumstances they are facing. Syrian refugees outside camps in Jordan scored higher than camp refugee residents of Oru community in West Africa (Oru-Ijebu, Southwest Nigeria (*Akinyemi et al., 2012*)) in all domains, except for the social relationships domain, which can be explained by the nature of the camp as it may provide a more social environment.

## Environment

Syrian refugees scored significantly higher than LSES Jordanians when asked about money availability; nonetheless they all scored below average. One-fifth of the registered Syrian refugees with UNHCR in Jordan receive cash assistance using iris-scanning biometric technology, to help them meet their basic needs (*United Nations High Commissioner for Refugees (UNHCR), 2017d*). In 2016, the Jordanian government has legalized work for Syrian refugees by giving work permits and encouraged the Syrian refugees to get legal work permits by easing the process. However, Syrian refugees were cautious due to application procedures and fear that their access to emergency aid would be at risk. The proportion of Syrian urban refugees who depend on humanitarian assistance as an income source, instead of work, dramatically increased in 2016. Only 36% of household earnings among Syrian urban refugees comes from work, and this percent continues to decline, as only 22% of them are working, compared to 35% of vulnerable Jordanians, in 2016 (*CARE, 2017*).

The unemployment rate was high in all groups, as being unemployed was around seven times more likely than being employed among the Syrian refugees, four times more among LSES Jordanians and two times more among ASES Jordanians. The employment

rate in Jordan was on average 32.95 between the years 2007 and 2018 which can explain the unemployment rate among Jordanian groups (*CARE, 2017*).

## Syrian urban refugees in Jordan and other refugees

Syrian refugees residing outside camps in Jordan scored lower than refugees in the camps of Gaza (*Eljedi et al., 2006*) and Kurdistan (*Aziz, Hutchinson & Maltby, 2014*) in the physical, psychological and social domains, and that may be due to the availability of free services inside the camp areas provided by the UNRWA and the UNHCR which offers a secure life for refugees on many aspects; food, health, shelter, and educational security, and this is not the case outside the camps. Shelter is the main concern for Syrian refugees outside camps in Jordan followed by utilities and education *United Nations High Commissioner for Refugees (UNHCR) (2017a)*.

## LIMITATIONS

This study was the first to assess the QOL of Syrian refugees residing outside camps, and recruited a large sample size, yet nonetheless has some limitations. First, recruiting LSES Jordanians was a challenging process due to the short permission time we had been given at SDD, and this resulted in a smaller sample size. Second, this study assessed the QOL of registered refugees, therefore, it was not representative of unregistered refugees, who were unable to get access to healthcare the way registered Syrian refugees do. Third, only a small portion of the population recruited answered the following question: ("How satisfied are you with your sex life?") in the social relationships domain, due to the sensitivity of the question in our cultural context. Another challenge faced in this study was that the males were less compliant when invited to participate in this study than females, which yielded in much less representation from the male perspective.

## FUTURE RECOMMENDATIONS AND DIRECTIONS

The results of this study emphasize the need for outreach and facilitation for psychological health support among both the Syrian refugees and LSES Jordanians. Additionally, Syrian refugees and LSES Jordanians need employment options. This is especially true with the nature of the long-term displacement the Syrian refugees are now experiencing. Providing job opportunities for both groups is the first step toward better psychological and environmental QOL. While all Syrian children are eligible for education, a lot of adults refugees are illiterate compared to Jordanians. In fact, illiteracy is 7% among Syrian refugees which is double that of our Jordanian sample. Better campaigns should target illiteracy to improve the overall QOL. Finally, there is a need for better fund outreach from external donor agencies to support the services provided to LSES Jordanians and Syrian refugees.

The Syrian crises did not only affect Syrians, it distressed other minorities who resided in Syria at the time of war. We recommend targeting other refugees affected by the Syrian crisis such as the Palestinian refugees registered originally in Syria but who fled to Jordan as a result of the crises.

## CONCLUSIONS

Average socioeconomic status Jordanians scored above the average in all domains and had the highest scores among all groups. Syrian refugees scored average or less in all domains, nonetheless, they scored significantly higher than LSES Jordanians in the environmental domain. LSES Jordanians had better psychological health and social relationships than Syrian refugees. No significant difference was found between Syrian refugees and LSES Jordanians in the physical health domain. Physical, psychological, and social domains were mainly affected by having a job, having higher income, and being married and free from diseases.

### Funding

This project was funded by the Deanship of Research at Jordan University of Science and Technology, Grant Number 20170218. The funders had no role in study design, data collection and analysis, decision to publish, or preparation of the manuscript.

### Grant Disclosure

The following grant information was disclosed by the authors:
Deanship of Research at Jordan University of Science and Technology: 20170218.

### Competing Interests

The authors declare that they have no competing interests.

### Author Contributions

- Nour Abdo conceived and designed the experiments, performed the experiments, analyzed the data, contributed reagents/materials/analysis tools, prepared figures and/or tables, authored or reviewed drafts of the paper, approved the final draft.
- Faten Sweidan conceived and designed the experiments, performed the experiments, prepared figures and/or tables, authored or reviewed drafts of the paper.
- Anwar Batieha conceived and designed the experiments, authored or reviewed drafts of the paper.

### Human Ethics

The following information was supplied relating to ethical approvals (i.e., approving body and any reference numbers):

The study protocol was approved by the institutional review board (IRB) of Jordan University of Science and Technology (JUST) IRB 33/107/2017.

### Data Availability

The raw data and SPSS code are available as Supplemental Files.

## Supplemental Information

Supplemental information for this article can be found online at http://dx.doi.org/10.7717/peerj.6454#supplemental-information.

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
