# Peer review of "Quality-of-life among Syrian refugees residing outside camps in Jordan relative to Jordanians and other countries"

_PeerJ, doi:10.7717/peerj.6454_

## Round 0.1 · original submission · Major Revisions

I recommend the authors revise the manuscript by including detailed explanations of the data and variables being used in this research (e.g., age and gender makeups of the sample data) and include discussions about the contributions and limitations of this research.

·

Basic reporting

Very important topic and potentially impactful research. Research question well defined,relevant & meaningful. It is stated how research fills an identified knowledge gap. Experimental design issue with comparing means in other countries. I think this should not be done and the authors should stick to presenting the rich data they already have form Jordan.

• A few typos found and corrections are needed. Highlighted areas that need word smithing and have grammatical issues. Please spell out WHOQOL-BREF in the abstract and all abbreviations at first mention in the text and abstract.
• Introduction: Suggest changing sentence in line 80 to: “and the majority of the remaining two millions are registered in Lebanon and Jordan.”
• Latest percentages from UNHCR Jordan show that more than 9% (Perhaps closer to 15%) reside in camps. Fix and include citation here. Also the 650,000 number is underestimated and numbers form the ministry of interior should be used (closer to 1 million refugees)
• 3 camps instead of two please check the literature. I believe there is a Dubai-sponsored camp or something.
• Line 85: suggest removing politically sensitive words such as “illegally”. One could debate that.
• Line 88: Again spell out QOL at first mention in the text. Abstract is not part of the manuscript.
• Include citation for QOL after end of sentence; line 91
• Line 93: The literature
• Line 126: Replace targeted with recruited
• Line 127: add a “total” before sample
• Line 142: spell out ASES
• Define criteria for ASES
• Lines 150, 151: Be very explicit in explaining the criteria for defining LSES
• Line 170: Too much wording. I suggest removing whole paragraph and briefly saying that it was validated with an included citation
• Line 174: What do you mean a pilot study was performed? What exact tool did you use to measure internal consistency? Did it measure within-subject variability or between-subject variability? Is the 0.7 a Cronbach’s alpha? If so, state that clearly.
• Lines 190&191. This is very low and there should be an explanation for that in the limitations section.

Experimental design

• Mention confounders you adjusted for in the multivariate regression model. Did you adjust for age? Confounders should be mentioned in this section not tin the results only. The statistical analysis section should be better explained to include variables used in the multiple regression model, adjustments made and comparison of means in other countries and refugee camps does not sound right. Comparing means is meaningless since context is different in each country and setting. How were the means in other studies obtained exactly? Through manuscripts review? Explicitly describe that. What was the n size per mean in each of these papers? An additional Table could be useful here stating source of information, authors, sample size, mean, year published..etc. I think comparing means is a weakness of this study and should not be part of the main findings but rather inserted in the discussion and throughout the general theme but does not sound right scientifically to compare means like this. If this was a systematic review paper this could be a different scenario and should be done vigorously well to show scientific rigor in presenting such data.
• Table 1: There should be an explanation in limitations why the majority of Jordanians recruited were females. Describe why unemployment rate in this sample especially among Jordanians is very high and how it relates to the national unemployment rate. For monthly income in table: include actual amount per category between brackets in JODs or US Dollars. What does presence of disease mean? Chronic condition or what? A common cold could be considered a disease at time of questionnaire administration.
• Social relationships: You mentioned earlier that the N size was small for this so how could you make such conclusions> Suggest minimizing generalizing like this with a small sample size.
• Line 253: again what diseases? You mean number of diseases? Or adjusting for having a disease or not?
• Table 4: what do you mean population as a variable here? Stratify by population and show data.
• Line 330: remove the word funding and use finances instead.
• Limitations section: what do you mean sample size was more than 200 so it is a good study? Reword. There is no set number for an ideal sample size. Remember this is a limitations section not a self-praising one. Again avoid charged terms such as “illegally”, you can say unregistered versus registered. I suggest removing the whole comparison with other countries and focus on the data in your paper that you collected and presenting it better in additional tables. You already have a lot of data. You can reference other papers in the discussion to try to draw comparisons but comparing non-camp to camp in other countries, lacking rigor in doing so is a major weakness. In the limitations you should mention why you had less answers for certain questions, why more females…etc. This is all faulty recruitment and study design that should be mentioned and accounted for.

Validity of the findings

Data is robust, statistically sound, & controlled. Stratify by population and present Table 4 in a better way. Remove comparisons with other countries. Suggest removing figures A & B

Additional comments

More citations and references should be included throughout the manuscript. Revise for typos and language

Reviewer 2 ·

Basic reporting

a. it is clear with professional English used throughout.
b. Introduction and background show context but did not mention the other refugees affected by the Syrian crises such Palestine refugees who were registered in Syria since 1948 and fled Syria to Jordan following the crises. According to UNRWA statistics a total number of 17,000 Palestine Refugee from Syria fled to Jordan as a result of the crises and are taken care by UNRWA.
c. Figures, tables and graphs are relevant.

Experimental design

a. The study is novel in assessing QOL among Syrian refugees in Jordan and clarifies the impact of crises and refugee status on QOL.
b. The data are statistically sound.

Validity of the findings

a. The research question is relevant, meaningful and identified knowledge gap.
b. The study is technically sound and ethically supported.

Additional comments

The study could have included other refugees affected by the Syria crises such the Palestine refugees who fled to Jordan which will widen scope of the study and add to its strength.
5. Conclusions are well stated and inked to research question; however it should recommend and open the window for new research with enough samples from all population affected by the Syria crises namely Syrian refugees, Palestine Refugees registered originally in Syria but fled to Jordan as a result of the crises, Palestine refugees registered in Jordan since 1948(total 2.3 million by Dec2017 as per UNRWA records in Jordan Field Office), Low socioeconomic status Jordanian, Average socioeconomic status Jordanian.
Attached is copy of the stud and annual report of health Department/UNRWA 2017.

Annotated reviews are not available for download in order to protect the identity of reviewers who chose to remain anonymous.

---

## Round 0.2 · accepted · Accept

Thanks again for your contribution to PeerJ. We look forward to receiving your high quality work in the future.

Reviewer 2 ·

Basic reporting

Professional article

Experimental design

Research question is clear

Validity of the findings

Novel and valid results

Additional comments

Well drafted